



# Mineral compounds in oak waterlogged archaeological wood and volcanic lake compartments

Giancarlo Sidoti[1], Federica Antonelli[2], Giulia Galotta[2], Maria Cristina Moscatelli[3], Davor Kržišnik[4], Vittorio Vinciguerra[3], Swati Tamantini[3], Rosita Marabottini[3], Natalia Macro[1], Manuela Romagnoli[3,*]

[1] Ministry of Culture, Central Institute for Restoration, ICR, Chemistry and material testing laboratory, Via di San Michele 25, 00153, Rome, Italy. giancarlo.sidoti@cultura.gov.it; natalia.macro89@gmail.com

[2] Ministry of Culture, Central Institute for Restoration, ICR, Biology laboratory, Via di San Michele 25, 00153, Rome, Italy. fedantonelli@gmail.com; giulia.galotta@cultura.gov.it

[3] University of Tuscia, Department for Innovation in Biological, Agro-food and Forest systems, DIBAF, S. Camillo de Lellis snc, 01100, Viterbo, Italy. mcm@unitus.it; marabottini@unitus.it; swati.tamantini@unitus.it

[4] University of Ljubljana, Biotechnical Faculty, Jamnikarjeva 101, SI-1000 Ljubljana, Slovenia. davor.krzisnik@bf.uni-lj.si, +386 (1) 320 36 15. davor.krzisnik@gmail.com

*Correspondence to*: M. Romagnoli (mroma@untius.it)

**Abstract.** Waterlogged archaeological wood (WAW) is a rare and precious organic material that can hold outstanding cultural values. In order to protect WAW for the next generations, this material must be accurately characterized to set its proper conservation, storage and exhibition conditions in museum environments. In this study, the mineral content found in WAW retrieved in a volcanic lake, was investigated by analysing wood ash through scanning electron microscopy coupled with energy dispersion spectroscopy (SEM-EDS). This micro-destructive approach was coupled with morphological studies carried out through optical microscopy. SEM-EDS was also performed on the WAW and on the surrounding sediment, to study the possible correlation between the mineral composition and the wood degradation state. The analysis revealed that calcium was the most abundant element in all poles with weight percentages ranging between 24% and 42%. This element was more represented in heartwood (HW) than sapwood (SW). In Sapwood the second most abundant element was arsenic. Sulphur, iron, and potassium were also present in all the analysed samples. Arsenic was detected also in the sediments; it was particularly concentrated in the samples taken near archaeological wood. The presence of this element can be linked to the volcanic origin of the lake, and its high concentration points to bioaccumulation processes induced by bacteria (erosion bacteria and sulphate-reducing bacteria) and biochemical processes favouring precipitation of insoluble compounds. The present work is the first investigation on mineral content in archaeological wood establishing a possible correlation with the surrounding volcanic lake sediments

**Keywords**: wood degradation, sediment, inorganics, arsenic, sulphur, calcium, iron.

## 1 Introduction

Waterlogged archaeological wood (WAW) can be defined as wood which was used by an extinct human culture and was then preserved in a water-saturated environment (e.g., lake, sea, rivers, land waterlogged sites) (EN16873, 2016; Rowell and Barbour, 1989). This precious material can overcome extended time of exposure in nature and be retrieved in a good preservation state thanks to particular conditions typical of waterlogged sites. The main factor contributing to wood preservation is the low oxygen concentration that slows down the microbial degradation by selecting anaerobic or microaerophilic microorganisms (mainly erosion bacteria and soft rot fungi) (Björdal, 2012; Nilsson and Björdal, 2009). Even if waterlogged wooden objects often preserve their original shape and dimension, the material undergoes chemical and physical modification that alter its features (such as changes in the cellulose/lignin ratio, augmented porosity, reduced density).

According to EN 16873 (2016), the physicochemical characterization of WAW is considered essential to set proper conservation protocols and to suggest the most appropriate environmental conditions for long-term storage and/or exhibition (Blanchette et al., 1994; Florian et al., 1990). Chemical characterisation of waterlogged wood is usually limited to qualitative and quantitative analysis of the main organic wood components (cellulose, hemicellulose, lignin) which most commonly deteriorate due to selective enzymatic hydrolysis by anaerobic microorganisms (Capretti et al., 2008; Macchioni et al., 2012; Pizzo et al., 2010; Romagnoli et al., 2018). On the other hand, physical analyses focus on properties related to wood mass and volume (e.g., maximum water content, basic density, residual density) (Babiński et al., 2014; High and Penkman, 2020; Jensen and Gregory, 2006; Macchioni et al., 2012; Pizzo et al., 2010).

Chemical and physical analyses are carried out almost routinely, while the content of inorganics inside the wood structure is an aspect often overlooked in the characterization of WAW. It is in fact well known that archaeological wood contains higher amounts of inorganics with respect to sound wood (Hedges, 1990); nonetheless, the process that leads to this accumulation is not still completely understood. Since microbial diversity inside archaeological wood (Landy et al., 2008), has been sometimes associated with the turnover of sulphur and iron (Björdal and Fors, 2019; Fors et al., 2014, 2012,



2008; Pop Ristova et al., 2017) the presence of some inorganic compounds could be associated to degradation by biotic processes. Sediments composition and elements present in water where the wood was preserved during centuries or millennia, influence greatly the content of inorganic compounds in the material. Furthermore, changes in wood porosity and permeability due to degradation processes affect the penetration of minerals. Usually, the concentration of metals increases as the loss of wood substances rises from cores to outwards (Broda and Frankowski, 2017). The accumulation of inorganics inside WAW can occur either by deposition on the surface of the cell walls or inside the lumens (process known as negative casting), or by replacement of the organic component (positive casting) and sometimes it can also occur in short times (Gillard et al., 1994). Accumulation of minerals in wood submerged under water or buried in soil in absence of oxygen, represents the first step of mineralization phenomenon, that can lead to fossilisation in geological times (hundred thousand or millions of years) (Fengel, 1991).

The possible influence of the laying time, and its extent in degradation processes and mineral accumulation in wood remnants remain currently mostly unquantified; however, it has been supposed that it plays a secondary role in the process (Kolář et al., 2014; Passialis, 1997) compared to the environmental conditions. In spite of the state of art where more aspects need to be clarified, among publications focused on quantifying the inorganic components in archaeological wooden objects very few analysed the relationships between these elements and the lying environment (represented by water and sediments) (Broda and Frankowski, 2017), and none, to the authors knowledge, focused on volcanic lakes.

Both sediments and water are important components of lake ecosystems and serve as sink and source of minerals and metallic ions. However, since these inorganic components can easily deposit and interact with sediments after entering water bodies, the final concentration of both minerals and metallic ions is expected to be higher in the sediments than in the water. On the other hand, when the physicochemical or hydrological conditions change, desorption and resuspension into the water body may occur. The distribution of minerals and metallic ions in water and sediments may in fact vary in relation to several factors either linked to their specific chemical behaviour (i.e., speciation) or to environmental properties. In particular, the accumulation of minerals from overlying water to the sediments may be influenced by the presence of organic and inorganic ligands, sediments type and available surfaces for adsorption, pH, electrical conductivity, redox conditions, temperature (Shalaby et al., 2017). Adsorption may be exerted by organic matter and/or the mineral component of the sediment. Particle size distribution affects heavy metals enrichment in sediments through physical and chemical processes (Huang et al., 2020). The geology of the basin also strong influences the content of minerals and metallic ions in lake sediments. Their distribution and relative concentration reflect, in fact, the occurrence and abundance of specific rocks or mineralised deposits in the drainage basin (de Anda et al., 2019). Cation exchange capacity of clays is a key factor in the adsorption of metals in aqueous solutions (Stathi et al., 2010).

Interactions of heavy metals with sediments' organic component or mineral fraction have been widely described in urban lakes (Yang et al., 2010) and rivers (Lin and Chen, 1998). Ferraz and Lourençlo (2000) reported that most of the metals were trapped in the sediment by dissolved organic matter through metal complexation.

Understanding the complex relationships with the surrounding environments, the set-up of possible phenomena of wood mineralization and in general the state and the rate of degradation of the material are key factors for the study of mineral compounds in waterlogged wood. Moreover, metals like iron, copper, and lead can interfere with WAW conservation causing depolymerisation of both wood components and preservation agents (e.g., consolidants like polyethylene glycol-PEG) (Broda and Frankowski, 2017); understanding and quantifying the minerals content is hence of paramount importance when choosing the more suitable restoration strategy for WAW.

The aim of this paper is to investigate the mineral content in archaeological wood found in a volcanic lake, comparing it with the surrounding sediment and water environment and establish a possible correlation between the mineral composition and the wood degradation state. To the knowledge of the authors, this is the first time that this type of investigation is carried out in a volcanic lake environment, and it is aimed to assess the best practice for conservation and understand wood modification process.

## 2 Materials and methods
### 2.1. Materials

The wood samples analysed in the present work were obtained from 8 oak poles belonging to pile dwellings of the Villanovan village of Gran Carro (about 8th century B.C.) in the lake Bolsena (Viterbo, Latium, Italy) which has been by long time object of archaeological studies (Fioravanti, 1994; Severi and Sciancalepore, 2016; Tamburini, 1995).

Cross-sections (ca.5-10 cm in thickness) were cut from the tops of the poles and named after the codes used by archaeologists to identify the poles (125, 126, 133, 144, 146, 152, 163, and 183). They were cut in cubical subsamples, following the orthotropic wood directions, to perform laboratory analyses. Specimens were prepared from both sapwood (SW) and heartwood (HW) of the same disc when present. Poles 126 and 163 represented an exception, having only heartwood and sapwood, respectively. For pole 163, analyses were carried out on two cubes taken from the external and the internal part of the section, named Sapwood outward (SW-out) and Sapwood inner (SW-in).



## 2.2 Methods
### 2.2.1 Wood

The degradation of waterlogged archaeological wood was measured by parameters well assessed in technical standard (i.e., maximum water content and ash content) while the organic fraction (cellulose, hemicellulose and lignin) was assessed by thermogravimetry. The results of physical and chemical characterization are reported in Romagnoli et al. (2018). The inorganic deposits noted inside wood structure during the micro-morphological characterization carried out during that study are the object of the present study.

At first, wood cubes were observed under stereomicroscope (Leica M205C). Then, thin sections (10–20 µm) were obtained for the three anatomical planes (cross, longitudinal-radial and longitudinal-tangential) by cryo-microtome (Cryostat CM 1900, Leica) and observed under optical microscope (Zeiss Axio Imager M2) using both bright field and polarised light. A deep insight by SEM was carried out on SW and HW samples showing a massive presence of inorganic material inside wood pores. Wood sections were attached to aluminium stubs using a carbon tape and sputter-coated with
gold in a Balzers MED 010 unit. The observations were made by a JEOL JSM 6010LA electron microscope using Secondary Electrons (SE) and Backscattered Electrons (BSE) detectors. To determine the elements, Energy Dispersive Spectroscopy (EDS) was also carried out.

Ashes obtained as a residue at 650°C from thermogravimetric analysis (TGA) (for procedure see Romagnoli et al., 2018) were examined without any treatment by scanning electron microscope Zeiss EVO 60 equipped with an INCA X-sight
dispersive X-ray spectrometer (EDS Oxford Instruments Detector 7636 Energy) for semi-quantitative analysis (SEM-EDS). Samples were analysed in variable pressure (100 Pa), working in backscattered mode. All the results are expressed as oxide percentage, excluding carbon and oxygen.

To further characterise the mineral components, X-ray diffraction (XRD) analysis was performed by Bruker D8 Advance diffractometer working with a Cu Kα radiation (40 kV and 30 mA) and a 1 mm beam collimator.

Regression analysis was performed between oxides percentages present in ashes and Maximum Water content as determined in Romagnoli et al. (2018). Furthermore, a comparative observation was carried out looking at the main elements found by sediment analysis and water chemical analysis.

### 2.2.2 Sediment and water

Sediment samples were collected in March 2022 in the vicinity of the 8 poles so to cover the entire study area (WSed).
In particular, the portion of sediment that surrounded each pole was carefully removed up to 10 cm depth and placed in a plastic vial for transportation. Four sub-replicates were collected from each pole. Additionally, to highlight sediment variations due to the presence of WAW, two areas outside the archaeological site, not interested by the presence of WAW, were selected as control and 10 sub-replicates were sampled in each area (Sed). A total of 52 sediment samples were collected and carried, stored with ice, in the laboratory.

Physical and chemical analyses of sediments included: texture, pH, redox potential (Eh), organic C (TOC) and total N content, total content of Al, As, Ca, Co, Cr, Cu, Mg, Na, Ni, P, Pb, S, Si, V, Zn W. Texture analysis was carried out in accordance with the Soil Survey Laboratory Methods Manual (Burt, 2014). Soil pH was measured potentiometrically in a 1:2.5 (w/v) soil-deionised water suspension with a pHmeter (Hanna Instrument) (van Reeuwijk, 2002) while Eh was assessed in a 1:5 (w/v) soil-deionised water suspension by means of a commercially available combination oxidation–
reduction potential (Redox (ORP)) electrode connected to a millivolt meter. (METTLER TOLEDO©, Italia).

Total organic C (TOC) and nitrogen content (TN) were determined by means of an elemental analyser vario-MACRO cube Elementar. Total metals content was assessed by means of ICP-OES.

Each sample was heated in an oven at 105°C for 48 h. The dried sediments to pass a 2 mm sieve. For each sample, three replicates of approximately 0.5 g each were placed into 100 mL PFA HP 500 Plus digestion vessel and 8 mL of ultrapure
(68% v/v) nitric acid and 2 mL of hydrogen peroxide (30% v/v) was added to each vessel, and then were loaded into a microwave oven (MARSXpress, CEM Corporation, Mathews, North Carolina). The digestion program (EPA Method 3052) was executed, and the samples were removed to cool. After cooling, the samples were filtered and diluted to a volume of 50 mL with milliQ water. For each batch of sample, reference blanks were prepared.

Mineralised sediment samples were subjected to measurement of the elements by emission spectrometry with ICP-OES
(Perkin Elmer® OptimaTM 8000 DV), in axial configuration and each sample was analysed in triplicate.

The elements determined and their corresponding wavelengths (nm) are listed as follows: Fe (259.939), Al (308.215), Pb (220.353), As (193.696), Cu (324.752), Zn (206.200), P (213.617), Ca (317.933), Mg (279.072), Cr (267.716), Co (228.616), V (292.464), Ni (231.604), Na (588.669), S (180.669), Si (SiO2)(251.611), W(224.876). Calibration curves for each of the elements listed were constructed with one calibration blank and nine calibration standards (standard



solution, CaPurAn, CPAchem, Stara Zagora, Bulgaria). To ensure that the instrument was performing consistently and efficiently the entire analysis an instrument performance check solution was analysed after every 15 samples and at the end of the analysis. The instrument performance check solutions consisted of solution at different concentration of the calibration standard.

Water samples were collected at poles depth, in 500 ml glass bottles at different sampling points covering the whole study area. Minerals and metals concentration was determined, by means of ICP-OES, using the same procedure as indicated for sediments. Water chemistry was comparable to the results obtained by Mosello et al (2004) showing an alkaline pH (8,2) and the highest concentration for Na (57737 ppb) followed by Mg (14425 ppb), Ca (10317 ppb) and S (586 ppb). P content was 9.0 ppb while, as for heavy metals, As was 25 ppb.

## 3 Results

### 3.1 Wood

Observations of all samples under stereomicroscope showed a diffuse presence of minerals inside wood pores appearing as a yellowish layer present in most parts of cells (Fig. 1a). In transversal thin sections this substance was opaque when observed in bright field (Fig. 1b), while it was characterised by orange birefringence under polarised light (Fig. 1c). The inorganic material was detected in cell lumina, often forming a layer adhering to vessels' secondary wall or filling

parenchyma cells.

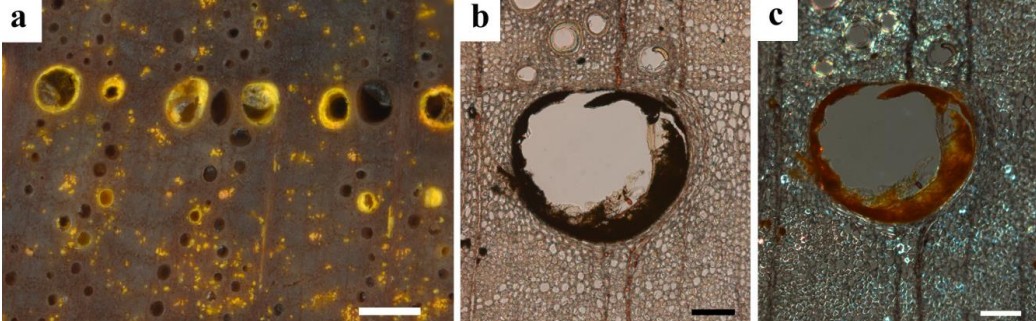

**Figure 1 - Pole 125, microscope view in cross section and elemental analysis of inorganic deposits in wood cells. (a) stereomicroscope; (b) bright field optical microscope; (c) polarized light optical microscope. Scale bar: (a) 0.5 mm; (b, c) 100 µm.**

Figure 2 shows sapwood BSE and EDS analyses. The yellowish layers present in fibres, tracheids and radial parenchyma are highlighted in white (Fig. 2 a,f) thanks to the BSE image, which means the yellow deposit were mainly composed of elements of high atomic weight. It seems that the internal cell wall of the vessel was covered in calcium and the yellow deposits were mainly composed of arsenic and sulphur. In particular, the two peaks of S and As trace back to the presence of arsenic sulphide compounds. The correct attribution to a specific mineral was carried out by means of XRD. The

analysis allowed identifying crystals of realgar ($As_4S_4$) (Fig. 3).



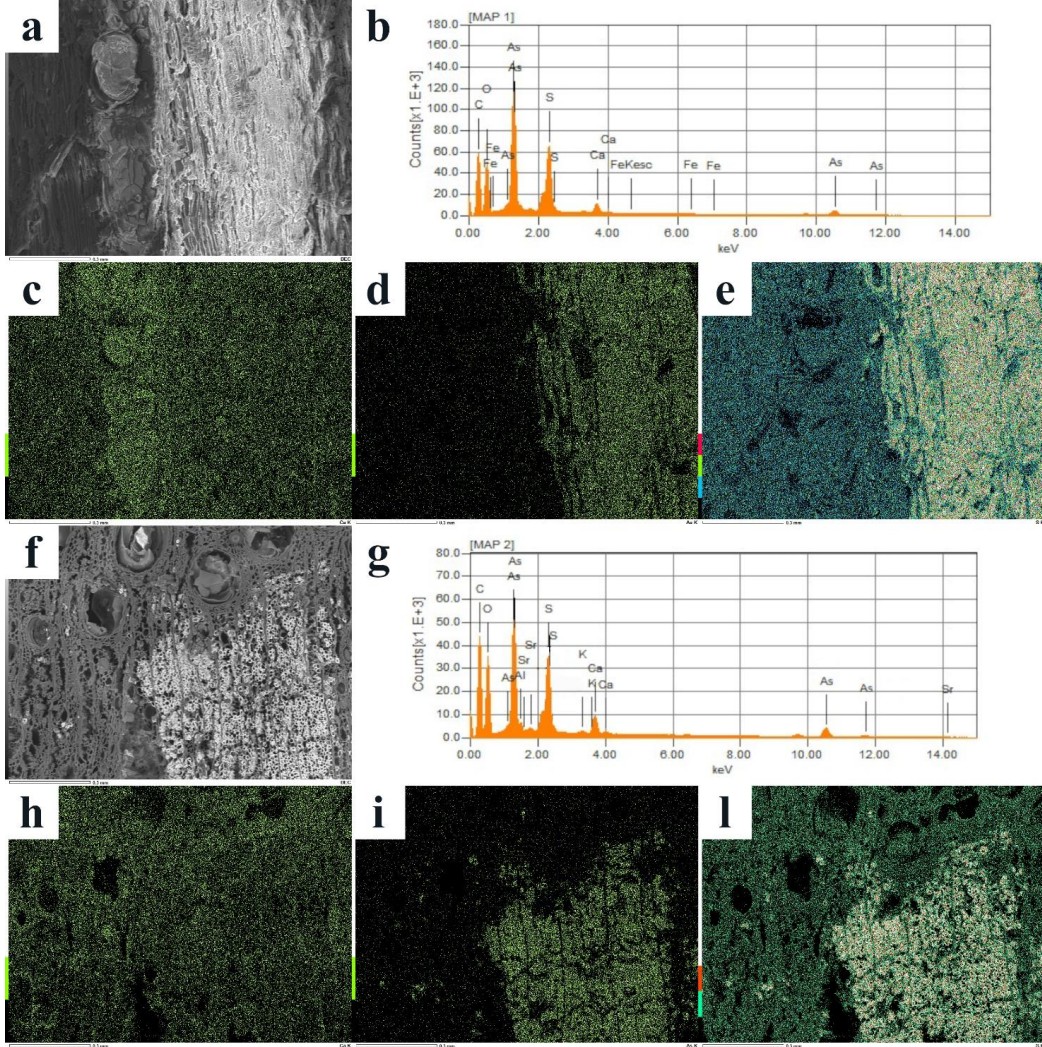

**Figure 2 - BSE (a, f) and EDS maps and charts (b-e, g-l) of sapwood representative samples. (a-e) Radial section. (f-l) Tangential section. (a, f) the white areas correspond to the yellowish deposit in the sample. (b) EDS chart of the section in Figure 2a. (c) Ca; (d) As; (e) S; (g) EDS chart of the section in Figure 2f. (h) Ca; (i) As; (l) S.**



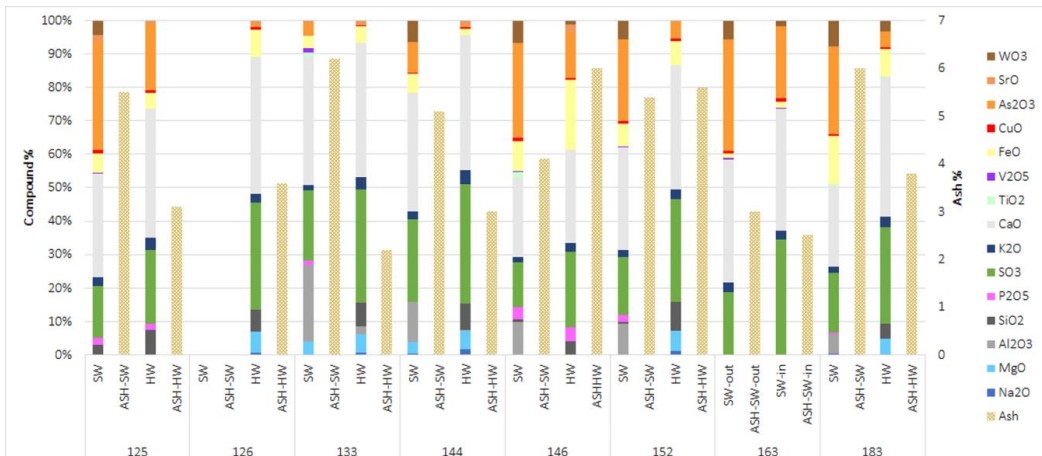

**Figure 3 – XRD analysis performed on yellowish minerals deposited in wood pores.**

SEM-EDS performed on ashes obtained from TGA highlighted the presence of 15 chemical elements. The weight percentage of elements reported as oxides and the ash percentage in the analysed wood samples are reported in the graph of figure 4 and in table S1.

As clearly shown, calcium was the most abundant element in all poles with weight percentages ranging between 24% and 42%. It was more represented in heartwood (HW) than sapwood (SW) (mean values SW 32%, HW 38%).

For SW, the second most abundant element was arsenic, present in all poles with a mean weight percentage of 22%. Its distribution was not homogeneous as shown by the variation coefficient of 82.5%. In HW, arsenic was even more variable. In particular, this element was absent in pole 133, present in traces (<1%) in poles 126 and 144, and exceeded 4% in 152, 183, and 146 reaching 20% in 125. Sulphur, iron, and potassium were also present in all the analysed samples with mean percentages 25%, 7%, and 2.7%, respectively. It is worth to emphasize that heartwood sulphur content was higher than sapwood for all the poles.

Other elements were detected only in a few poles. Aluminium and tungsten were always present with higher values in sapwood, on the opposite magnesium and silicon were more represented in heartwood. Other elements like sodium, phosphorus, titanium, vanadium, copper, and strontium were present with percentages lower than 4%, without a differential distribution between SW and HW.

**Figure 4 - SEM-EDS analyses of ashes obtained from TGA. Elements are given as oxides (%). Ashes reported as % on wood dry weight.**



To evaluate the effect of the most abundant metals on the wood state of preservation, correlations between metal percentages and the Maximum Water Content (MWC) were investigated (Fig. 5). The analyses showed that no linear regression was present between As, S and heartwood MWC, while for Ca a negative correlation was observed ($R^2$=0.76).

For sapwood, the association between the two variables was around 0.5 for Ca ($R^2$=0.49), and As ($R^2$=0.54), and 0.86 for S. In the case of Ca and S the correlation was positive, this means that a greater amount of these elements was present in the most degraded wood. For As the correlation was negative, the highest As values were related to the less deteriorated material.

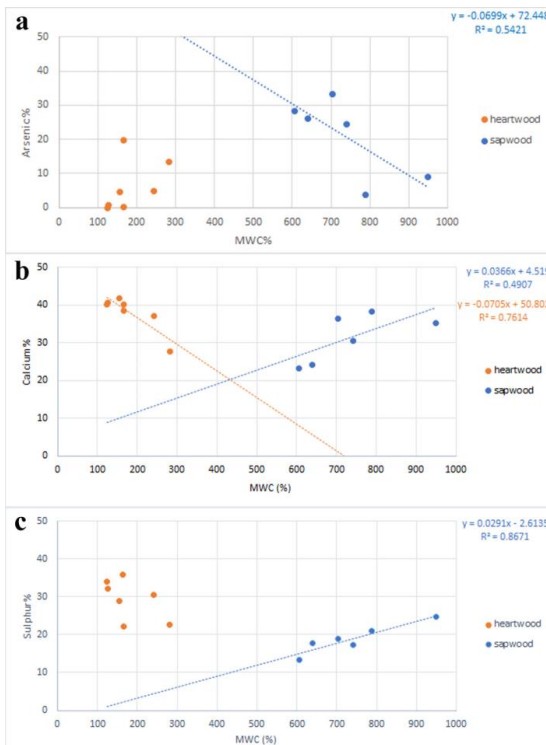

**Figure 5 – Regression analyses performed between the percentages of the three most abundant metals and Maximum Water Content (MWC). (a) Arsenic; (b) Calcium; (c) Sulphur.**

### 3.2 Sediments

The physico-chemical properties of Bolsena lake sediments are reported in Table 1. Sediments texture was sandy, loamy-sand, the pH was neutral/slightly alkaline with a low significant acidification in WSed, total organic carbon ranged from 0.17 to 21.9% while total nitrogen from 0.02 to 1.16 %; TOC was significantly higher in WSed. The redox potential indicated anaerobic reducing conditions and significantly increased in WSed showing enhanced availability of $O_2$.

The minerals and heavy metals concentration in the sediments of Bolsena lake are reported in figures 6a, b and c. For almost all heavy metals, P, S and Si the content in WSed was always significantly higher than in Sed indicating a specific concentration of these elements in the vicinity of the archaeological wood (Fig. 6b and c). Conversely, for minerals such as Fe, Al, Na, Ca and Mg their abundance was significantly higher, except for Al, in the sediment not in contact with wood (Sed) (Fig.6 a, b).

Table 2 summarises the "wood effect" representing the percentage variation of each mineral/metal concentration in WSed with respect to Sed. Fe, Al, Mg, Na, and V show a negative effect of the wood presence while for all the other heavy metals a significant positive variation has been observed in WSed.

| | Texture | TOC | TN | pH | Eh |
|---|---|---|---|---|---|
| | | % | % | | mV |
| Sed | Sand, Loamy-sand | 1.3±0.3 | 0.3±0.1 | 7.4±0.05 | -36.3±1.9 |
| WSed | | 8.6±2.0 | 0.5±0.1 | 7.1±0.10 | -30.0±1.6 |




| P value | | *** | ns | *** | ** |
|---|---|---|---|---|---|

**Table 1. Main physico-chemical properties of Bolsena lake sediments. Sed: sediment, WSed, sediment surrounding waterlogged wood. TOC: total organic carbon, TN: total nitrogen, Eh: redox potential. Standard error is reported. P value represents the**
**level of significance between the average of Sed and WSed samples (One way Anova, Tukey post hoc test): *: p<0.05, ** p<0.01, *** p<0.001.**

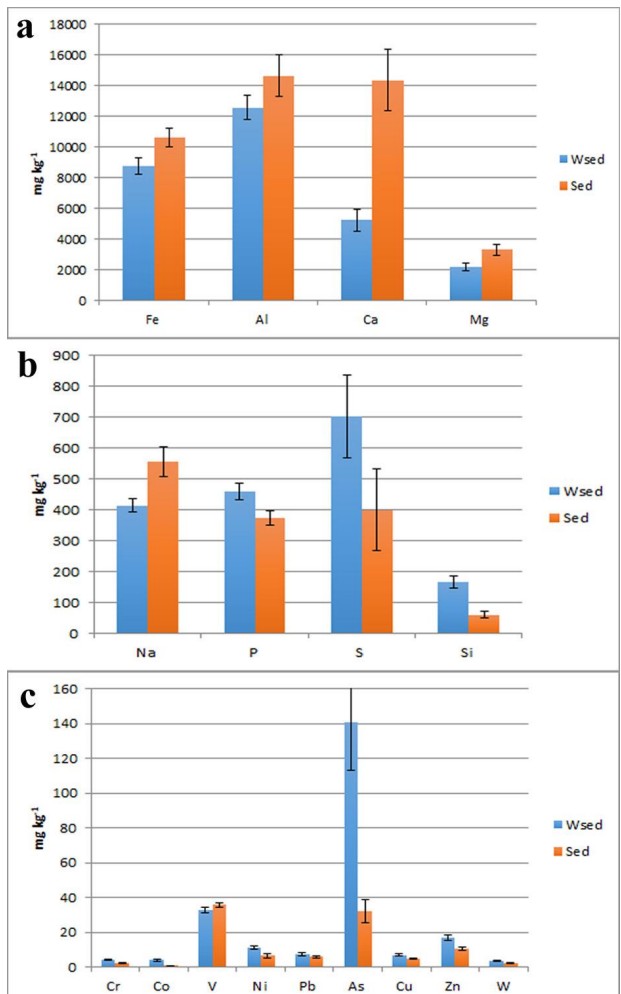

**Figure 6 – Concentration of minerals and heavy metals in the sediments of Bolsena lake. (a) Elements originated from the lithological substrate; (b) elements of mixed origin; (c) Heavy metals (natural and/or anthropogenic sources).**

| | Al | As | Ca | Co | Cr | Cu | Fe | Mg | Na | Ni | P | Pb | S | Si | V | W | Zn |
|---|---|---|---|---|---|---|---|---|---|---|---|---|---|---|---|---|---|
| **%** | -14 | +337 | -64 | +338 | +100 | +43 | +17 | -17 | -33 | -25 | +70 | +23 | +24 | +76 | +177 | -8 | +51 | +56 |
| **P value** | ns | * | *** | *** | ** | * | * | * | ** | ** | ns | ns | ns | *** | ns | * | ** |

**Table 2: Percentage variation of sediment minerals and metallic ions content due to wood presence. Positive values indicate increase of metal concentration due to wood presence. P value represents the level of significance between the average of Sed and WSed samples (One way Anova, Tukey post hoc test): *: p<0.05, ** p<0.01, *** p<0.001.**

## 4 Discussion and conclusions

In the discussion of the results the interactions between different biological and environmental aspects must be considered.
Particularly relevant are the following factors:  the concentration of minerals in standing tree (specifically oak wood), the specific laying conditions (i.e., type of elements and their concentration in sediments), and the level of degradation of waterlogged archaeological wood.



The chemical composition of recent oak wood is characterized by holocellulose (over 40% on average) and lignin, ranging from 19% to 27% in percentage (Kolář et al., 2014; Krutul et al., 2010). Ash content, which represents the mineral
component of the material, in recent wood is in general lower than 1% (Baar et al., 2020; Fengel and Wegener, 1984; Kolář et al., 2014). WAW degradation impacts wood mineral content. The study by Romagnoli et al. (2018) showed that the heartwood of the poles from the Gran Carro village is quite well preserved, with medium-low values of MWC. On the other hand, the corresponding sapwood was always more degraded reaching MWC values over 900%. In all the analysed poles, the amount of ash was, on average, 4.3%, (maximum 6.2%). These values exceed 1% which is usually
found in recent wood but are below 10-13%, which is the mean value in WAW (Fengel, 1991; Hedges, 1990; Lucejko et al., 2020; Passialis, 1997; Tsoumis, 1969). The increased percentage of ash in degraded waterlogged wood is due to the loss of the organic components (cellulose, hemicellulose, and lignin) and the relative enrichment in minerals that are not removed by biodeteriogens (Hedges, 1990). Furthermore, the biodegradation of cell wall results in increased porosity (Han et al., 2020; Svedström et al., 2012), favouring accumulation of minerals in the tissues. In wood remnants of lake
Bolsena, the most degraded sapwood is more susceptible to the accumulations of inorganics as testified by the percentage of ashes always higher compared to the corresponding heartwood (Fig. 4, Table S1), indeed there are differences also in the composition of minerals present in both tissues because some of them are more represented in heartwood than in sapwood. In term of ash composition, it is interesting to note that the most abundant elements usually present in recent oak wood were found also in the WAW of lake Bolsena. In recent oak, calcium is the most abundant, followed by
magnesium and potassium. In particular, the latter is a metal fundamental for plant photosynthesis and respiration (Broda and Frankowski, 2017; Schiopu and Tiruta-Barna, 2012). Moreover, Mn, Na, P, Cl, Al, Fe, Zn, Cu, Pb, Co and Si are reported in trace in the ash of the recent wood (Baar et al., 2020; Kolář et al., 2014; Tsoumis, 1991).

Bolsena lake lies within the Vulsino volcanic district in which the volcanic activity occurred in alternate phases. The geological evolution of the district is complex showing the presence of vulcanites (derived from lava and ignimbrite
flows) and a clayey flyschioid sedimentary substrate (Mosello et al., 2004). The mineral composition of sediments is strictly related to that of lake water and is strongly dependent on geological substrate nature and weathering processes, atmospheric deposition, biological processes, physical properties (such as pH, Eh, temperature etc.). Also, for very mobile and particularly reactive elements, a seasonal fluctuation can be observed, and it is due to the temperature changes, occurrence of overturn. The concentration of elements found in Bolsena sediments shows the following ranking Al> Fe>
Ca> Mg> S>P>Si. A large concentration of Ca, Mg and S was also found in the water chemistry at Bolsena Lake by Mosello et al (2004). Ca was the most abundant inorganic element also in WAW and usually it was higher in heartwood than in sapwood, with the lowest variability in both the tissues compared to the other minerals. Waterlogged wood is degraded by bacteria which cause a pH decrease, confirmed in this study in WSed. This may suggest larger mobilization of Ca from calcium carbonate forms and can be one of the reasons for a lower amount of Ca in sapwood compared to
heartwood. Ca is very well represented in standing trees and recent wood as well (Broda and Frankowski, 2017).

Concerning the presence of heavy metals in sediments, the highest concentration was found for As followed by V, Zn, Ni, Pb, Cu, Cr, W and Co. However, despite As, all heavy metals concentrations did not exceed the PEC (probable effect concentration) as reported in the sediment quality guideline (MacDonald et al., 2000) indicating, therefore, the lack of potential toxicity within this environmental matrix in the lake of Bolsena. Conversely, in the sediment surrounding the
waterlogged archaeological wood As content reached 140 mg kg$^{-1}$ as average value (the maximum value was about 500 mg kg$^{-1}$), exceeding thus the PEC of 33 mg kg$^{-1}$ (MacDonald et al., 2000). However, As concentration in the lake water was very low (25 µg L-1) being 150 µg L-1 the threshold of chronic pollution (Ingersoll and Mac Donald, 2002; Osuna-Martínez et al., 2021).

A significant "wood effect" was therefore found in this study for As, Co, Cr, Cu, Ni, Pb and W indicating a preferential
concentration of these heavy metals in the sediment surrounding the WAW. This effect was particularly evident also for other elements such as P, S and Si.

The reasons of this peculiar behaviour should be found in the different environmental conditions characterizing the WSed. Shaheen et al. (2020) reported decreasing or increasing concentrations of heavy metals, in lake sediments, in response to a gradient of pH and Eh applied during a 22 days incubation in microcosms. In the present study, in WSed, the different
environmental conditions such as the increase of Eh values accompanied by a lower pH, could have induced an immobilization of heavy metals favouring their speciation and/or ionization that may have promoted adsorption processes on sediment and consequently into WAW structure.

In particular, this mechanism may explain the specific biogeochemistry of arsenic in this environment. The presence of As in sediments and then in WAW is obviously linked to the volcanic origin of the lake (Samadzadeh Yazdi and
Khodadadi Darban, 2010). The high concentration of this element in the sediment points to its immobilization process through a potential adsorption/fixation either on sediment mineral and/or organic fraction or on wood tissues. Arsenic chemistry in anaerobic environments depends on pH, Eh and on Fe oxidation state. At the environmental conditions characterizing Bolsena lake sediments, such as neutral/subalkaline pH and Eh ranging from -45 to -10mV, the dominant As species is $H_3AsO_3$ (AsIII, mobile and toxic form). Furthermore Fe-oxides minerals, that can adsorb the oxidized form





(AsV) in these conditions, release the metal as AsIII when Fe is turned from FeIII to FeII further mobilizing therefore the metal. The high level of As in wood and wood-surrounding sediments can be thus explained by bioaccumulation processes induced by bacteria and biochemical processes favouring precipitation of insoluble compounds. In micro-aerophilic and near anaerobic conditions the lignocellulosic structure of wood is actively broken down by erosion bacteria (EB). The simple sugars produced by this process represent a metabolic source for sulphate-reducing bacteria (SRB), a group of
anaerobic prokaryotes classified as secondary wood degraders or scavenging bacteria, able to reduce sulphate, $SO_4^{2-}$, to sulphide, $S^{2-}$, during the metabolization of simple organic molecules (Fors et al., 2008). The hydrogen sulphide produced by these microorganisms in the presence of Fe and As ions is not stable and tends to transform into iron and arsenic sulphides that precipitate in wood porous structure (Fors et al., 2012; Sandström et al., 2003). The precipitation of As in the form of realgar, one of the insoluble forms of As, and its further accumulation into WAW was confirmed in this study
through the XRD analysis (Fig, 3). This could explain the significant negative relationship between As and SRB found in this study (r= -0.36, p<0.01; data not shown). In a recent review, Sun et al. (2016) showed that SRBs can use arsenic to form insoluble sulphide mineral-like phases in the form of an orpiment-like phase ($As_2S_3$), a realgar-like phase (AsS), an arsenopyrite-like phase (FeAsS) (Alam and McPhedran, 2019; Fors et al., 2012; Sandström et al., 2003). As previously hypothesized, the metabolism of SRB is strictly involved in As transformations under anaerobic environments. SRB
depend on sulphate availability as a source of energy. In this study the activity of arylsulfatase, the enzyme involved in the release of sulphate from organic matter, was positively and significantly related to SRB biomass (data not shown).

Fors and colleagues (2008) proved that in waterlogged wood EB and SRB promote the accumulation of sulphur, as thiols in the lignin, and iron sulphides, as particles, and that a positive correlation exists between these elements and bacterial wood degradation. A further study (Björdal and Fors, 2019) demonstrated that an analogous role can be played by soft
rot fungi that, degrading cellulose and hemicellulose components of cell walls, and creating a substrate valuable for secondary colonisers, actually promote the accumulation of both compounds, particularly in the rich in tannins oak wood.

In the present work, the correlation ($R^2$=0.86) existing between S and MWC (Fig. 5c) confirms the relationship between this element and wood biodegradation in sapwood. In fact, the graph clearly shows that the highest S concentrations are reported in the most degraded samples. This same correlation was not observed in heartwood samples (Fig. 5c) that, on
the other hand, were always less degraded and had higher S percentages with respect to sapwood. A similar pattern of degradation and S distribution was observed by Björdal and Fors (2019) in oak samples from the XVII century shipwreck called the Crown. The authors hypothesised that the accumulation of S in the deeper, more anoxic, and more preserved parts of the wood can take place via chemical reactions, without the presence of microbial degradation. Anyway, this hypothesis has still to be investigated to provide further explanations.

As for arsenic, in sapwood there seems to be a negative correlation between concentration and degradation. The highest values of As are found in the best preserved samples. This could be explained by assuming an initial phase of wood bacterial degradation which led to the production of hydrogen sulphides and then to the precipitation of As. The presence of the latter subsequently slowed the progression of biological degradation. This effect is not perceptible in heartwood in which the presence of As is not related to the level of degradation which is much lower than that of sapwood. The natural
durability of oak heartwood must have prevailed over the preservative effect of this element.

Regarding the presence of iron in WAW, it must be taken into account that, regardless of bacterial attack, oak wood is particularly susceptible to iron accumulation due to the high concentration of tannins that form stable chemical compounds reacting with iron so that natural wood colour turns into dark brownish to almost black (Broda and Frankowski, 2017). This could explain the highest concentration of Fe in heartwood where tannins are mainly present.

The presence of tungsten in wood is probably related to the volcanic origin of the lake. In fact, W in the form of calcium and iron tungstate has been reported in the volcanic area of the Tyrrhenian side of the Italian peninsula (Bellatreccia et al., 1999). As already discussed for As, tungsten is a redox-sensitive element exhibiting contrasting geochemical behaviours under different redox and/or sulfidic conditions (Johannesson et al., 2013; Watanabe et al., 2017). Therefore, it is possible that in the environmental conditions characterising Bolsena lake also this metal may be subjected to
immobilization processes in the sediment and then in the WAW. Also, silicon is ascribable to the volcanic nature of the lakebed. Its non-uniform distribution in wood samples is probably linked to the random penetration of sand grains inside wood structure (Fors et al., 2008). Furthermore, it must be considered that Si is usually found in heartwood of recent oak (Kolář et al., 2014).

Total P content in sediments was on average about 400 mg kg$^{-1}$. Zhang et al. (2021) indicate values of total phosphorus
below 450 mg kg$^{-1}$ as index of nominal P pollution. Low contents of P were also found in the water lake in accordance with the results reported in Mosello et al. (2004) and in the availability of sediment labile P forms (data not shown) pointing, therefore, to the absence of an eutrophication process. Studying chemical modification of wood induced by bacteria, Gelbrich and colleagues (2008) proved that in WAW the content of phosphorus increases for higher degrees of bacterial attack. In the present work no significant correlation was observed between state of preservation and P content.
In fact, the highest P percentage was recorded for heartwood of pole 146 (4.4%) that had a low-medium level of



degradation (mean MWC 280%) while this element was absent in the worst preserved poles (e.g., 144S with MWC>900%).

The presence of elements found in low concentration in the archaeological wood (e.g., K, Al, Mg, Na, Cu, V) is more difficult to explain taking into account external influences as they are usually reported also in recent wood as components of organic macromolecules or involved in cellular metabolic reactions (Baar et al., 2020; Broda and Frankowski, 2017; Tsoumis, 1969). Aluminium, present mainly in sapwood, sometimes has been explained as the presence of aluminium-containing minerals (e.g., clay minerals) in the soil.

**Funding sources.** This work was supported by JPICH-19 – Italian Ministry of University project "Archaeological Wooden Pile-Dwelling in Mediterranean European lakes: strategies for their exploitation, monitoring and conservation (WOODPDLAKE)". Funding source had no involvement in study design; in the collection, analysis and interpretation of data; in the writing of the report; and in the decision to submit the article for publication.

**Author contribution.** Conceptualization: Sidoti, Galotta, Moscatelli, Antonelli, Romagnoli. Methodology: Sidoti, Romagnoli, Galotta, Moscatelli, Kržišnik. Investigation: Soditi, Marabottini, Kržišnik, Tamantini. Writing - Original Draft: Romagnoli, Galotta, Antonelli, Moscatelli. Writing - Review & Editing: Romagnoli, Galotta, Antonelli, Moscatelli Tamantini. Project administration: Romagnoli and Ciabattoni. Funding acquisition: Romagnoli. All authors have read and agreed to the published version of the manuscript.

**Acknowledgements.** The authors would like to thank Barbara Barbaro, archaeologist of the Superintendence of Archaeology, Fine Arts and Landscape for the metropolitan area of Rome the Province of Viterbo and Southern Etruria together with the technician Egidio Saveri, the staff of the Underwater Archaeology Research Centre (CRAS) of Bolsena and Marco Ciabattoni, conservation scientist of ICR, for their support in underwater sampling and monitoring of lake environmental parameters.

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
