# Peer review of "Inorganic component in oak waterlogged archaeological wood and volcanic lake compartments"

_EGUsphere, 2022_

## Author Comment (AC3)

The manuscript "Mineral compounds in oak waterlogged archaeological wood and volcanic lake compartments" describes interesting research on the relationship between the elemental composition of waterlogged archaeological wood and the environment where it was buried for a long time. The study is important from the conservation perspective since the presence of various minerals can hinder proper conservation treatments or contribute to further wood degradation.

In general, the manuscript is well-written, the experiments well-planned and conducted, and the results presented clearly and understandably. However, the paper lacks conclusions and I would suggest supplementing this part carefully, considering both the conservation perspective and the environmental aspect.

**Citation**: https://doi.org/10.5194/egusphere-2022-1498-RC1

We added the conclusions

**General comments**

The manuscript "Mineral compounds in oak waterlogged archaeological wood and volcanic lake components" provides a good starting point for understanding the mechanisms behind the degradation of waterlogged archaeological wood and its chemical interaction with its surrounding substrate. I found this an extremely interesting read with greater applications to the realm of archaeology. The study is well structured with relevant methods/analyses used.

However, the major issue is the use of EDS, which is fundamental to the study. The authors do not document the methods behind their analysis well enough for me to judge if the data they acquired is accurate. Settings, such as the acceleration voltage used, surface topography, acquisition method (maps vs spot scans) etc should be documented as standard practice, as they can severely impact the quality and error within the data sets (Newbury DE, Ritchie NWM 2013, Shirley & Jarochowska 2022). This is not to say by any means that the chemical composition recorded here is incorrect, but without providing this information, it cannot be evaluated fully or reproduced.

Without the inclusion of this information I would suggest this paper should be reconsidered after revisions and the EDS data properly evaluated.

**Specific comments**

The manuscript refers to mineral content and composition throughout, including the title, but EDS does not provide mineral composition; it provides elemental composition and, under some strict assumptions, relative concentrations. Methods that provide mineral composition are e.g. XRD or EBSD. Mineral phases are also not reported in the manuscript so this needs to be corrected throughout.

The text and title were corrected.

Line 115 to 120: Clarification needs to be made on the parameters of the EDS analysis as the description is lacking. All user defined parameters have an impact on the data acquired and I would like to see some information.

1) The acceleration voltage that was used should be documented. This has a direct impact on the reliability of the spectra acquired (Ie Figure 2 b and g - these are spectra, not "charts" and it should be indicated where (spot, line, area) they were taken). The general rule of thumb is that the acceleration voltage (kV) used should be twice that of the activation voltage (keV) of your element. So if we presume that the authors used 20 kV in their analysis, then only elements in the range of 0-10 keV provide reliable results. My gut feeling is the authors may have used 15 kV, which is fine, it just needs to be clarified.

Line 997. We added the information requested (20 kV) and we changed the terminology in Figure 2 caption.

2) The authors state that the samples were coated in gold (at least the thin sections). Was EDS conducted on these samples while they were coated? If so, the activation voltage of Au Kβ is 2.12 keV which, on a gold coated sample, may have some negative impacts on elements with similar activation voltages like P Lα at 2.013 keV, S Lα at 2.37 keV. Modern systems can adjust for this in post processing and also using higher kV can be really useful, but without the information above, it's hard to say what has been done.

Since we analysed woody archaeological artifact, which could collapse, there was a need of cover. We have experience with gold, which is a common method in SEM observation, notwithstanding we knew the possible interferences there might be in the analysis. In any case, the distribution of gold is homogeneous, on the opposite, S and K map correspond with eachother, meaning there was not interference.

3) Other information, such as if the system was standardized to another material (Newbury and Ritchie 2015), dwell time, working distance are also desirable.

The text was modified as requested, details on EDS analyses were provided. EDS analyse s were performed at 20 kV, 12 mm of working distance, with 100 ⬜s of dwell time, on an area of about 4 mm$^2$.
The analysis on wood were made with a standardless SEM instrument.

Line 125: Was the Ash prepared before going into the SEM? It says there was no treatment.

Ashes were not gold or carbon coated, we explicitly clarified this point in the text (line 124). Wood samples were gold sputter-coated as reported in line 119.

Are there BSE images? How were they mounted and were they coated? Also there are (generally) three different forms of EDX measurements 1) Spot scan 2) Line scan and 3) Maps (What I presume is used in Figure 2). What was used here? Each of the above methods has its benefits and pitfalls, with spotscans being generally accepted as providing the best quantitative results. However, Figure 2 leads me to believe that the spectra acquired in panel b and g were acquired from the map data? I think this is fine for this specific use case, but was the same method used for the data from ash? While okay for relative concentrations of elements, topography can have some serious impacts on the quality of data acquired (Shirley & Jarochowska 2022). I would need to see more information on how the data was acquired from the ash samples before commenting on its validity. Some SEM images with the area in which the data was acquired, is one way of achieving this.

Yes, as explained in Figure 2 caption. For wood analysis we explained the used method in lines 994-997. We added the information about the EDS measurement method (mapping). For the ashes…

Line 127: How were the values normalized? Most systems use the software that is provided with the EDS installation, so I'm presuming because this is Oxford Instruments either Inca or Aztec were used for this calculation. While both are "black box" for these calculations, the complexity of how their numbers are calculated is beyond the scope of the paper here, I would just like the software to be named.

The Inca 5.05 software was used to collect and analyse data, with a cobalt quant optimization. These details were reported in the text    .

It should be noted that the values were acquired under variable pressure (100 kPa is very low) so there will be a lot of interference from residual air and this makes the measurements semi-quantitative at best.

As reported in the original text, pressure was 100 Pa, not 100 kPa. Furthermore, there is no residual air in the chamber but nitrogen. The measurements made were semi-quantitative, we had already underlined this point at line 126.

Line 194: I don't think I have access to the supplementary data, but it would be necessary to follow some details, e.g. the use of "variation coefficient" of 82.5% for arsenic (l. 198), which seems very high; do the authors mean coefficient of variation or perhaps just the variance?

The value reported is the coefficient of variation not the variance. The high value is due to the fact that it was obtained considering both values from sapwood and heartwood. CV% for As in sapwood in 47.2%, in heartwood is 121.5%. We thank the reviewer for his comment and catch this opportunity to clarify this point in the text.

Lines 203-206: mean values in the text would be useful to support the text, e.g. when differences between SW and HW are mentioned, what are the actual mean values for individual elements in these two wood types?

The reviewer is right, the mean values help in understanding data. However, adding mean sapwood and heartwood mean values for each element would make text difficult to read. So, we decided to add all means in the supplementary table. We will submit revised supplementary material together with the revised manuscript.

Lines 210-215: Were P values also calculated for this regression analysis?.

- "correlations were investigated" - no correlations are reported in the manuscript; perhaps you mean "relationships"? "negative correlation was observed (R2=0.76)." - R2 is not a correlation coefficient and cannot be negative - please sort out the use of correlation/relationship/association throughout the text. Also do mention this is least squares regression, because that's what Excel calculates.

You are right, we changed the terminology. The negative relationship can be seen by curve trend.

- Line 211: What do you mean that "no linear regression was present"? I presume that the authors deemed that a R2 value below a threshold meant no relationship? But what was that threshold? A coefficient of determination can be calculated for any set of points; the text should probably say "no relationship was present".

- Six points seems like very little, do the authors have any more they could work with? Or take multiple measurements from an individual cite on a sample?

The reviewer is right, we will modify the text substituting "correlation" with "relationship". Unfortunately, in studying archaeological artifacts it is always difficult to have a wide statistical sample because of the limits in sampling a low number of samples. We have no more data available to work with.

Figure 2: I think this figure needs a bit of work before publication. While the overall layout seems fine there are some suggestions to clarify.

- The scale bars in the bottom left corner of each map need to be bigger, in the current version it is very hard to read. They appear to all be the same scale, so one would do for the whole figure. Same for the element names on the bottom right of each panel, either make them bigger or remove them.

We changed the bars and tags accordingly.

- Should figure 2 "l" not be a "j"?

We changed the letter (we counted in Italian).

- Panels c,d,e,h,i,l all have very small and hard to read scale bars with no readable scale to represent the number of counts.

Panels changed.

o e has a different color scheme. Are these maps even comparable?

Yes, they are comparable, the color is different just because the intensity of the signal is higher.

- Figure 2 b and g: the y axis label is a bit confusing and the overall text too small.

Changed

o First I suggest keeping the measurements as whole numbers (20.0 counts to just 20).

Done.

o Secondly I suggest a change is made from Counts [x1.E+3] to "Counts [x1e+3]","Counts (Thousands)", or "Counts x103".

Done.

o Similar to the y axis, the x axis should have larger text and maybe drop the decimal points because here they are unnecessary.

Done.

o Why is there a label to Map 1 and Map 2?

They refer to the different mapping (Map 1 is for figure a, c-e, and Map 2 is for figure f, h-j), we deleted it, since it looks confusing.

o Panel g What is FeKesc?

It means the x-rays photons escaped (esc) the detectors over the K orbital (K) in the case of iron element (Fe). In practice, there is a statistical probability that some of the x-rays, emitted from the EM sample can interact with Fe K-shell electrons in the detector. That is, for a photon of the x-rays with energy higher than the Fe K absorption edge, the photon can knock out some Fe K-shell electrons in the detector and involves a process of secondary x-ray generation (generating Fe K x-ray) in the detector itself. There will not be an impact on the detected signal if this process still contributes to the overall charge collected for the original incident x-ray photon from the EM sample. However, there is a probability that the generated Fe Kα x-rays escape from the detector volume and do not contribute to the charge collected for the original photon that is detected. In our case we have different peaks for Fe, so we can sum them in order to have the total charge collected by the detector.

Lines 220-224: the text does not agree with Table 1. "total organic carbon ranged from 0.17 to 21.9% while total nitrogen from 0.02 to 1.16 %" - these are different values than reported in Table 1. Is this because Table 1 reports mean or median values? Then it should be made clear and the number of measurements per each category should be shown. Yes, Table 1 reports mean values. of Wsed and Sed samples. The n. of replicates per sample was 32 replicates for WSed and 20 for Sed. Statistical analysis was performed using JMP11.0 (SAS Inst, Inc. Cary, NC, USA) software package. Caption has been modified reporting the above information.

Tables 1 and 2: "P value represents the level of significance" - no, P value represents the chance of obtaining such a difference by chance, in the absence of differences between compared groups, if the assumptions of the test are met. Levels of test significance are chosen by the users. Caption has been modified

What software were the tests carried out in? Indicate the numbers of individual samples in each group and (related) the degrees of freedom. The n. of replicates per sample was 32 replicates for WSed and 20 for Sed. Degrees of freedom ($n_1 + n_2 - 2$) = 50. Statistical analysis was performed using JMP11.0 (SAS Inst, Inc. Cary, NC, USA) software package.

What does ns stand for? Table 1 and Caption was changed, P values were reported in the tables. Ns stands for not significant and has been clarified in the caption

Perhaps it would be helpful to spell out the null hypotheses that are tested in both cases and discuss them as such in the text. In similar studies the null hypothesis ($H_0$) is that the two samples are extracted from the same population while the alternative hypothesis ($H_1$) is that the two samples belong to different populations.

**Technical corrections**

Figure 1 - Well composed and easy to understand figure. It would be nice to be able to link panel b/c are to the panel a with a box eg. (I believe on the right of panel a correlates to b/c?).

We are grateful to the reviewer for his kind comment. Even if taken from the same sample, it is not possible to link b/c to a point in section a because panels b/c are thin sections (not only magnifications of what observed in panel a) and do not actually correspond to a point of panel a.

Line 55 - The sentences starting with "Sediments composition" and "Furthermore, changes in wood" would benefit from some references.

References were added in the text.

Figure 3 needs editing: subscript should be used in oxide formulas where appropriate. What are the "in" and "out" suffixes in the X axis labels?

We thank the reviewer for the suggestion. The image was modified using subscripts. The suffixes "in" and "out" refer respectively to the inner and outer part of sapwood of pole 163, as reported in materials and methods at line 107.

Figure 5: improve readability by changing MWC to Maximum Water Contents in the X axis label. R2 values are too small to read. Are the line equations used for anything in this study? If not, leave them out.

Line 110 - Should (WAW) be used in place of waterlogged archaeological wood here?

The text was modified as requested.

Line 120 - What is the thickness of the sputter coating? And was EDS conducted on the sputter coated sample? The thickness of a gold coating can affect the acquisition of chemical data so it's important to just note that here.

It is 8 ± 1 nm. We added the information in methods paragraph.

Line 125 - was gold also excluded? I'm guessing the ash was not gold coated, but it's not really clear in the methods.

Ashes were not gold or carbon coated, we explicitly clarified this point in the text (line 124).

Line 180: There may be charging of the surface here, it would be good if the authors could just clarify if this is the case or not.

Line 182: Great! Double checking with XRD is a good idea.

Line 287: L-1 should be L-1

Done.

Line 334: A bit colloquial when using Anyway here

We modified the text.

Line 361: what is pole 144S and how does it differ from pole 144?

We are sorry for this little typo, we intended 144 SW. The text was corrected.

Minor language issues, e.g. "The geology of the basin also strong influences the content of minerals and metallic ions in lake sediments." (l. 79-80), should be e.g. "The geology of the basin also influences the content of minerals and metallic ions in lake sediments strongly." (adverb, not adjective); similar issues elsewhere in the text

Citation: https://doi.org/10.5194/egusphere-2022-1498-RC2

---

## Author Response (AR1)

Dear editor,

We considered the referee's comment about the correlation of the different parameter, therefore we changed the term "correlation" in "relationship".

We corrected some typos.

We changed figure 4, 5 and 6 as they were not suitable for persons with colour vision deficiencies.

We removed the equation in figure 5 as requested by the referee.

We modified table 1, so that the header includes measurement unit and therefore we removed the second line.

We rearranged the conclusion paragraph.

---

## Author Response (AR2)

**Answer to Associate Editor**

**L. 26:** biogeochemical not biochemical
Done.

**L. 42, 45:** On the other hand always goes with on the other. If not then use, However.
If we correctly understood the comment, we modified the sentence accordingly to the associate editor's request.

**L. 54:** Sediment
Done.

**L. 67:** delete lying
Done.

**L. 72:** However,
Done.

**L. 135:** sediment and water chemical analysis
Done.

**L. 142:** replace interested with impacted
Done.

**L. 144:** collected and transferred on ice to the laboratory.
Done.

**L. 153:** L. 159: Sediment samples..
We were not sure about what the comment meant since there are 2 "sediment sample" in the same line. Anyway, we changed both, please check if it is fine by your side.

**L. 166:** efficiently during the entire analysis, an instrument…
Done.

**L. 167:** solutions with
Done.

**L. 173**: use . systematically for decimal not ,
Done.

**L. 174:** reformulate: while, as for heavy metals, As was 25 ppb. Unclear to reader.
The sentence was rewritten.

**L. ?** L. 259: However,
We could not find the correction you requested.

**L. 289:** L. 290: However, except As, all heavy metals
Done.

**L. 294:** L. 295: 150 microg L-1 being the threshold
Done.

**L. 314:** L. 315: biogeochemical
Done.

**L. 382:** biogeochemical
Done.